# Dynamics of Phase Synchronization between Solar Polar Magnetic Fields Assessed with Van Der Pol and Kuramoto Models

**DOI:** 10.3390/e22090945

**Published:** 2020-08-27

**Authors:** Anton Savostianov, Alexander Shapoval, Mikhail Shnirman

**Affiliations:** 1Gran Sasso Science Institute, viale F. Crispi 7, 67100 L’Aquila, Italy; 2National Research University Higher School of Economics, Myasnitsakaya Ulitsa 20, 101000 Moscow, Russia; ashapoval@hse.ru; 3Institute of Earthquake Prediction Theory and Mathematical Geophysics, Russian Academy of Sciences, Profsoyuznaya Ulitsa 84/32, 117997 Moscow, Russia; shnir@mitp.ru

**Keywords:** solar hemispheres, solar faculae, synchronisation, coupled oscillators

## Abstract

We establish the similarity in two model-based reconstructions of the coupling between the polar magnetic fields of the Sun represented by the solar faculae time series. The reconstructions are inferred from the pair of the coupled oscillators modelled with the Van der Pol and Kuramoto equations. They are associated with the substantial simplification of solar dynamo models and, respectively, a simple ad hoc model reproducing the phenomenon of synchronization. While the polar fields are synchronized, both of the reconstruction procedures restore couplings, which attain moderate values and follow each other rather accurately as the functions of time. We also estimate the evolution of the phase difference between the polar fields and claim that they tend to move apart more quickly than approach each other.

## 1. Introduction

Scientists are challenged by existed regular patterns and examples of irregularities in the dynamics of the hemispheric asymmetry in the solar magnetic field [1,2]. The asymmetry in the amplitude is investigated numerically (see, f. e., [3]) and implemented into the flux transport dynamo models [2,4]. The hemispheres contribute non-equally into the solar activity; their dominance is changed inside various solar cycles including 12–15, 17–18, and 21–23 [5,6] The evolution of the phase difference exhibits its own patterns. The hemispheres are typically synchronized [7,8,9], and the phase-leadership of one of them is persistent within several cycles with only two changes in the leadership occurred near 1928 and 1968 during the previous century [10,11].

The northern and southern components of the solar magnetic field can be connected through meridional circulation which transports the field from the equator towards the poles and backward forming the cells [12]. Regular observations of the meridional circulation have been reported only recently. Their empirical investigation and explanation with the flux transport models reveal general mechanisms but still leave open questions regarding the number and geometry of the cells, the influence of the meridional circulation to the solar cycle, and the dynamics of the velocity profile of the meridional circulation [13,14,15,16,17,18].

We are motivated by Blanter et al. [11] who investigated the asymmetry between the hemispheres with the Kuramoto model of the coupling oscillators by reconstructing the dynamics of the phase difference between them. This approach complements other applications of the inverse problem widely exploited to estimate properties of the deep solar dynamo processes that cannot be observed or measured at the solar surface [19,20]. Despite the fact that the Kuramoto model is not derived from the first principles, the range of its applicability includes laser arrays, neural networks, chemical oscillators, and neural activity [21,22,23,24]. Ad hoc modeling, simplifying the underlying physical processes and capturing the effect of synchronization, can give an accurate solution of the inverse problem in just mentioned fields. The choice of an appropriate model is always a challenge. Savostianov et al. [25] linked the efficient reconstruction of the “hidden” features of the solar activity discovered with the Kuramoto model to analogous reconstructions performed with the Van der Pol model of the coupled oscillators, where the latter is considered as a (very) simple descendant of the MHD equations (after [26,27]).

The purpose of this paper is to introduce a measure of similarity in the reconstructions of the coupling performed with the Kuramoto and Van der Pol system of two oscillators. The developed theory is applied to the solar faculae time series, which represent the northern and southern polar fields, to investigate the phase asymmetry in the coupling between the solar hemispheres. Finally, we plan to restore the phase difference between the time series that results in the found level of the similarity in the coupling.

This paper develops our previous study which reveals the similarities in the reconstructions of the coupling between the International Sunspot Numbers ISSN and geomagnetic index aa inferred from the Kuramoto and Van der Pol models of the coupling oscillators [25]. These reconstructions are built on the assumption that the coefficients of the corresponding differential equations vary slowly; in other words, the properties of the solutions are inherited from the equations with the constant coefficients. Now we address the question regarding the dynamics of the coupling and the phase difference when estimating the proximity of the reconstructions.

The applicability of the theory of coupled oscillators to the polar faculae data is worth discussing. A strong correlation between the two time series is observed with a “naked eye”. This correlation is a consequence of the mutual coupling which exists as the chain of the following interactions. A poloidal magnetic field associated with a pole and represented by the solar faculae transits to the near-equator toroidal field. The transition occurs within one of the hemispheres. Further, the poloidal field of this hemisphere is connected to that of the other hemisphere through cross-equatorial links established because of the different polarities of the magnetic fields. Finally, the poloidal field of the other hemisphere is transformed into the toroidal field. The reverse coupling occurs in a symmetric way. Probably, this complex mechanism “trying” to synchronize the polar faculae represents the coupling of the solar hemispheres. As to the transition between the poloidal and toroidal fields, it is performed with the meridional circulation analyzed in empirical way, with ad-hoc models, and in the framework of the solar dynamo modeling [14,28,29,30].

The rest of the paper is organized in the following way. Section 2 introduces the data. Section 3 describes the method of the analysis introducing the two reconstructions. The main findings are placed in Section 4. Section 5 concludes.

## 2. Data

We use yearly MWO calibrated polar faculae data downloaded from the solar dynamo dataverse (https://dataverse.harvard.edu/dataverse/solardynamo), maintained by A. Munoz-Jaramillo [31]. The faculae data representing the corresponding solar polar magnetic fields share the common solar quasi-periodicity of ∼11 years with the solar cycle.

Despite the distant locations of the data sources, the faculae time series illustrate a strong phase synchronisation (Figure 1). Nevertheless, from time to time the series are de-synchronized. The most evident episode of de-synchronization occurred during cycle 19 in the 1960s. We investigate the synchronization of the series, exploring their dynamics in sliding windows and applying general models which efficiently describe synchronization.

## 3. Method: Reconstruction of the Coupling with Two Models

Here we describe the Kuramoto and Van der Pol models of two coupled oscillators. The inverse problem intended to reconstruct the coupling between the oscillators is formulated and solved, given the other parameters of the equations and their solution.

### 3.1. Kuramoto Model

In the case of the Kuramoto model, the oscillators are given by the system of the equations
(1)$$\begin{array}{c} x\left(t\right)=\sin\theta_{x}\left(t\right) \end{array}$$
(2)$$\begin{array}{c} y\left(t\right)=\sin\theta_{y}\left(t\right), \end{array}$$ where the dynamics of the phases is determined by the coupling coefficient μ and their natural frequencies $\omega_{x}$ and $\omega_{y}$ through the differential equations
(3)$$\begin{array}{c} {\dot{\theta }}_{x}=\omega_{x}-\mu \sin\left(\theta_{x}-\theta_{y}\right) \end{array}$$
(4)$$\begin{array}{c} {\dot{\theta }}_{y}=\omega_{y}-\mu \sin\left(\theta_{y}-\theta_{x}\right). \end{array}$$

Simplifying comparison between different models, the time normalization is used in such a way that 1 is assigned to the synchronized natural frequency: $(\omega_{x}+\omega_{y})/2=1$. We introduce half a difference Δω between the natural frequencies:$$\Delta \omega =\frac{\omega_{x}-\omega_{y}}{2}.$$

Thus, the direct problem is the description of the solutions $\theta_{x}$, $\theta_{y}$ of Equations (Equation 3) and (Equation 4), given Δω and μ.

We also consider the inverse problem for the Kuramoto model formulated in the following way: given series x(t), y(t), which satisfy Equations (Equation 1)–(Equation 4), and the model parameter Δω, reconstruct the coupling coefficient μ between the oscillators. Following Blanter et al. [32], the solution of the inverse problem is inferred from the computation of the correlation between the free oscillators. We define the sliding correlation in the window of the length *T* as:(5)$$\mathrm{Corr}\left[X,Y\right]\left(t;T\right)=\frac{E_{t,T}\left(\left(X-E_{t,T}X\right)\left(Y-E_{t,T}Y\right)\right)}{\sqrt{E_{t,T}{\left(X-E_{t,T}X\right)}^{2}E_{t,T}{\left(Y-E_{t,T}Y\right)}^{2}}},$$ where $E_{t,T}X=T^{-1}\int_{t-T/2}^{t+T/2}X\left(\tau \right)d\tau$, and put it into the correspondence to the center of this window. One can expect the relationship between the coupling and the correlation. The larger coupling is the stronger it causes the oscillators to follow each other and exhibit bigger values ρ(t)=Corr[X,Y](t;T) of the correlation. When the oscillators are synchronized, so that $\dot{\theta_{x}}-\dot{\theta_{y}}=0$, Equations (Equation 3) and (Equation 4) yield
(6)$$\mu \sin(\theta_{x}-\theta_{y})=\Delta \omega .$$

The condition μ>Δω provides the synchronization. Let T=2π be fixed to the period of the synchronized oscillators. Then the correlation between x(t) and y(t) computed with (Equation 5), where x(t), y(t) are substituted for X(t), Y(t) is given (see [32] for the proof, which consists of the straightforward computation of the integrals) by the equation
(7)$$\rho =\cos(\theta_{x}-\theta_{y})$$
with the autonomous right hand side. Equations (Equation 6) and (Equation 7) lead to the following reconstruction rule:(8)$$\mu =\frac{\Delta \omega }{\sqrt{1-\rho^{2}}}$$

Since the right hand side of (Equation 8) equals to $\sin\arccos(\theta_{x}-\theta_{y})$ up to the sign, this sign describes the leadership among the two oscillators. Assigning only “+” as the sign, one eliminates the information about the leadership. Equation (Equation 8) also implies the following simple relation between the correlation and the coupling: the higher the coupling between two oscillators, the more correlated they are (thus, the smaller stationary phase difference they exhibit).

We are going to apply the reconstruction rule (Equation 8) when x(t) and y(t) represent observed oscillators. We stress that solar proxies are not fully synchronized, in contrast to the solutions of the model equations. Nevertheless, Equation (Equation 8) is formally applicable, as soon as Δω is given (the choice of Δω will be discussed later). Since the correlation between solar proxies varies with time (ρ=ρ(t)) it follows that the coupling reconstructed with the rule (Equation 8) is also time-dependent (μ=μ(t)). The usage of Equation (Equation 8) is plausible if the genuine coupling (i.e., that is between solar proxies) varies slowly with time:(9)$$\frac{d}{dt}\left(\theta_{x}-\theta_{y}\right)\approx 0.$$

Equation (Equation 9) exhibit quasi-synchronization. The assumption of quasi-synchronization underlies our approach. Clearly, if the coupling does vary slowly, then the quasi-synchronization regime is quickly attained. The waiting time depends only on the distance from the initial condition to the attractor.

### 3.2. Van Der Pol Model

The pair of the Van der Pol (VdP) coupled oscillators is governed by the equations
(10)$$\begin{array}{c} \ddot{x}-\left(1-x^{2}\right)\dot{x}+\left(1-\Delta \omega \right)x+\mu \left(\dot{x}-\dot{y}\right)=0 \end{array}$$
(11)$$\begin{array}{c} \ddot{y}-\left(1-y^{2}\right)\dot{y}+\left(1+\Delta \omega \right)y+\mu \left(\dot{y}-\dot{x}\right)=0, \end{array}$$ where μ is the symmetrical coupling of the oscillators and the natural frequencies 1±Δω are introduced in such a way that the common synchronized frequency, i. e., their half a sum, is normalized to 1. The normalization is in line with that for the Kuramoto oscillators. Kuznetsov et al. [33] and Astakhov et al. [34] found similar characteristics of the direct problem for systems (Equation 3), (Equation 4) and (Equation 10), (Equation 11), whereas Savostianov et al. [25] discussed similarities of the inverse problems posed for these systems.

The formulation of the inverse problem coincides with that for the Kuramoto model. However, the solution is obtained numerically. Namely, Equations (Equation 10) and (Equation 11) are solved for a broad domain of the pairs (Δω,μ) and the correlation $\rho =\rho_{\mathrm{VDP}}\left(\Delta \omega ,\mu \right)$ of the solutions is computed with the window of the length *T*, where *T* is the period of the limit cycle of the synchronized oscillators. In numerical computations, we wait for a sufficiently long time prior to the data processing to avoid the influence of the initial conditions on the correlation. Figure 2 illustrates the relationship $\rho_{\mathrm{VDP}}$ between ρ, Δω, and μ. The reconstruction of the coupling μ is given by the inverse function
(12)$$\rho_{\mathrm{VDP}}^{-1}\left(\Delta \omega ,\rho \right)$$
also found numerically. The inverse function is well defined if the correlation ρ is positive and |Δω|<μ. Performing numerical computation, we assign 0 to $\rho^{-1}\left(\Delta \omega ,\rho \right)$ when the correlation is negative.

Function $\rho_{\mathrm{VDP}}^{-1}\left(\Delta \omega ,\rho \right)$ shown in Figure 2 and the relationship described by Equation (Equation 8) have much in common. According to them, when the coupling strengthens or the phase difference shrinks, the oscillators evolve more alike each other and their correlation becomes bigger. A fall in the coupling strength causes a drop in the correlation (Figure 2, upward changes along vertical lines with fixed values of Δω).

### 3.3. Reconstruction Scheme

The Kuramoto and VdP models give the two ways of the reconstruction of the coupling exhibited by (Equation 8) and (Equation 12) respectively. The dynamics of the reconstruction is inferred from the time series x(t) and y(t) considered as the input when restoring coupling under the assumption of the quasi-stationarity of the solutions. We add into the consideration the dynamics of the differential equations by generating a new series. The full reconstruction procedure is displayed in Figure 3. It consists of the following steps.

(A)Given time series X(t), Y(t), and the model parameter Δω, we reconstruct the coupling with both models. These series exhibit the solar cycle; $T^{*}=11$ years is used in the paper to assign a *single* number to the *variable* length of the cycle. The time axis is initially stretched by $T^{*}/2\pi$ to transform the estimate of the cycle length into 2π and set the correspondence between the time axis in the models and observations. Clearly, the linear transform does not affect either the correlation between the series X(t) and Y(t) or the variability of the solar cycle. The Kuramoto reconstruction is performed with (Equation 8) and denoted $\mu_{k}\left(t\right)$. The VdP reconstruction is performed with (Equation 12) and denoted $\mu_{v}\left(t\right)$. These two procedures are schematically displayed in the left two blocks of Figure 3. The both reconstructions $\mu_{k}\left(t\right)$ and $\mu_{v}\left(t\right)$, in general, depend on time, since the input series represent the observations instead of the solutions of the model equations. The mathematical expectation of the input series is switched into the mean when the correlation is computed.(B)Following Equation (Equation 7), we put
(13)$$\theta \left(t\right)=\arcsin\frac{\Delta \omega }{\mu (t)},\;x=\sin\left(t+\theta \left(t\right)/2\right),\;y=\sin\left(t-\theta \left(t\right)/2\right)$$
and define the correlation of the series ρ(t)=Corr[x,y](t;T) over the period, where $\rho =\rho_{k}$, $\theta =\theta_{k}$, $\mu =\mu_{k}$, $x=x_{k}$, and $y=y_{k}$ when the Kuramoto model is investigated. The subscript *v* is used for the VdP model. The series x(t) and y(t) introduced by (Equation 13) represent free oscillators (the second block from the right in Figure 3).(C)Finally, we repeat the reconstruction of the coupling from the time series and the phase difference (the first block from the right in Figure 3). Equation (Equation 8) is applied for the both types of the input to get ${\hat{\mu }}_{k}$ and ${\hat{\mu }}_{v}$ from $\rho_{k}$ and $\rho_{v}$ respectively; Δω has been fixed during the steps (A)–(C). This part involves the dynamics of the equations into the reconstruction. Namely, the addressed question is how the dynamics of the coupling in the direct problem affects reconstruction. We end up with the reverse transform of the time axis and restore years as the units of the reconstructions found in the paper and displayed on the Figures.

The described procedure greatly benefits from the normalization applied in the modeling. Instead of tracking initial physical variables from the MHD equations (which also implies the existence of uncertainties when choosing the values of the parameters), we turn to the normalized quantities, thus shrinking the parameter space down to the coupling μ and the normalized frequency difference Δω. Baring that in mind, the coupling strength μ should be treated as a composite of various physical quantities related to the interactions of the oscillators.

### 3.4. Comparison of the Reconstructions

In this section, we discuss the proximity between the reconstructions ${\hat{\mu }}_{k}$ and ${\hat{\mu }}_{v}$ to each other. By construction, both reconstructions are located above Δω. Let
(14)$$r_{\Delta \omega }\left[{\hat{\mu }}_{k},{\hat{\mu }}_{v}\right]\left(t\right)=\frac{{\hat{\mu }}_{k}\left(t\right)-\Delta \omega }{{\hat{\mu }}_{v}\left(t\right)-\Delta \omega }.$$

This ratio is expected to exhibit larger variations than ${\hat{\mu }}_{k}\left(t\right)/{\hat{\mu }}_{v}\left(t\right)$, as both reconstructions can drop to a neighborhood of Δω. The stability of $r_{\Delta \omega }$ as a function of time and, moreover, the proximity to 1 give evidence that the reconstructions with the Kuramoto and VdP models agree with each other.

The ratio $r_{\Delta \omega }$ is well defined at the points *t* such that the correlation ρ(t) obtained at step (A) is positive and *t* belongs to the time interval offset by the period *T* from both ends of the considered time span. The two-time usage of the sliding windows of the length *T* generates the requirement about offset. The inverse function $\rho_{\mathrm{VDP}}^{-1}$ computed through Figure 2 is well posed if the correlation is positive.

We note that there are other quantities besides the pair $\left({\hat{\mu }}_{k}\left(t\right),{\hat{\mu }}_{v}\left(t\right)\right)$ that are worth investigating when discussing the reconstruction of the coupling. The proximity of the initial reconstructions $\mu_{k}\left(t\right),\mu_{v}\left(t\right)$ obtained at the step (A) also evidences in favor of the interchangeability of both models. The transition from ${\hat{\mu }}_{*}\left(t\right)$ to $\mu_{*}\left(t\right)$, where the star stands either for *k* or *v*, leads to more smooth curves offset to the right (see [35]). Therefore, we’ve decided to restrict themselves here only to the comparison of the quantities $\left({\hat{\mu }}_{k}\left(t\right),{\hat{\mu }}_{v}\left(t\right)\right)$ obtained at the end of the reconstruction procedure.

## 4. Results

### 4.1. Reconstructed Couplings and Relation between Two Models

Interpreting Δω, one can say that the “period” of the quasi-periodic series X(t) and Y(t) belongs to the interval $\left[T^{*}-\Delta \omega T^{*},T^{*}+\Delta \omega T^{*}\right]$, where $T^{*}$ is a point estimation of the solar cycle period. The setting Δω=0.2 defines a feasible range of the values attained by the period.

The reconstructions ${\hat{\mu }}_{k}\left(t\right)$ and ${\hat{\mu }}_{v}\left(t\right)$ of the coupling with Δω=0.2 are displayed on the top panel of Figure 4. The pink vertical stripes indicate the time span where the reconstruction is not well defined. The computation of the correlation is performed with a smaller window when *t* belongs to the left and right pink time stripes. The middle pink time stripe corresponds to the values which are obtained a negative correlation between the series. In this case, $\mu_{v}$ is set to Δω, which is the least feasible value of the coupling. The two curves ${\hat{\mu }}_{k}\left(t\right)$ and ${\hat{\mu }}_{v}\left(t\right)$ follow each other rather accurately (Figure 4, upper panel).

The consistence between these curves conserves when the ratio $r_{\Delta \omega }\left(t\right)$ defined by (Equation 14) is considered, Figure 4, lower panel. Outside 1958–1975 interval, the ratio $r_{\Delta \omega }\left(t\right)$ varies slowly attaining values inside the range 0.83–0.91, Δω=0.2 (black curve). Changes of Δω within the interval [0.1,0.3] slightly affect the average value of $r_{\Delta \omega }\left(t\right)$ but not the weak variability of $r_{\Delta \omega }\left(t\right)$, Figure 4 (lower panel).

### 4.2. Reconstruction of the Frequencies

Now we are going to find a variable frequency Δω(t) which generates a fixed value of $r_{\Delta \omega }\left(t\right)$ which is attained most frequently. To this end, the range of $r_{\Delta \omega }\left(t\right)$, reachable at each time moment *t* with all possible values of Δω, is found (numerically). This range is located between the two curves shown in Figure 5. The values within the light red corridor are attained outside the interval 1958–1982. The width of the interval with attainable values of $r_{\Delta \omega }\left(t\right)$ varies with time remaining almost unchangeable after 1980.

The above analysis supports the conjecture that the two reconstructions of the coupling are similar in terms of the ratio $r_{\Delta \omega }\left(t\right)$. More precisely, this $r_{\Delta \omega }\left(t\right)$ demonstrates a typical range of the values, given fixed Δω from a broad interval (0.025,0.5), assuming a uniform distribution of Δω and focusing on the relevant values.

Then one can assess the phase difference between the oscillators assuming that $r_{\Delta \omega }\left(t\right)$ is constant, but Δω(t) varies. We choose 0.86 as a “typical” value of $r_{\Delta \omega }\left(t\right)$ and restore Δω(t) which results in this $r_{\Delta \omega }\left(t\right)$. The evolution of the restored 1+Δω(t) interpreted as the normalized phase frequency associated with the northern component of the polar magnetic field is shown in Figure 6. Small values of Δω in the late 1950s and around 1916 are observed (Figure 6) approximately 7–10 years prior to the changes in the leadership of the oscillators representing the northern and southern solar hemispheres (see Blanter et al. [11] and Deng et al. [36] for the discussion regarding the changes in the leadership and the time of their occurrence). The leadership of the northern hemisphere were ceased to exist in the 1920s and recreated in the 1960s. The second change is known to be much stronger. Since each restored value Δω(t) is obtained with the 22-years data centered at this *t*, these small values of Δω are constructed with the data which represent the changes in leadership. Our rough way to restore the natural frequencies failed at the time of this strong change in the leadership and resumed its functionality in the mid 1980s when the phase difference returned to usual values (Figure 6). A downward trend at the right part of the graph in Figure 6 may also attain values of the 1920s and 1960s, but with the data at hand only the level of the 1940s is reached.

The reconstructed dynamics of the natural frequencies is asymmetrical: the intervals characterized by the outward drift of the two hemispheres (around 1915, 1945, 2000) are noticeably shorter than the intervals with the phases moving to each other (e.g., 1920–1940, 1945–1960, 1990–2000), Figure 6. In general, this asymmetry is not novel and difficult to model. Following Syukuya and Kusano [37], one can attribute the asymmetry to the interactions between dipole- and quadrupole-type solution of the MHD model focused on the properties of the two hemispheres. With this model, Syukuya and Kusano [37] generate the slow phase convergence and quick divergence between the hemispheres (observed in Figure 6) when the corresponding solutions approaches the attractor.

Moreover, the MHD dynamo models admit various patterns in the long-term dynamics of the asymmetry, not limited by phase differences. Passos et al. [38] constructed the range in the feasible dynamics of the N–S asymmetry, including the grand-minima of only one of the hemispheres and almost complete synchronization as possible extreme cases. A few examples of the asymmetry in the phase-difference dynamics derived implicitly in our paper need further empirical justifications.

## 5. Conclusions

The paper compares the two reconstructions of the coupling between the polar faculae which represent the polar components of the solar magnetic field. The reconstructions are inferred from the Kuramoto and VdP models of the pair of coupled oscillators. Kuznetsov et al. [33] derived the Kuramoto-like equation for phases from the VdP equations Savostianov et al. [25] the established similarity of the inverse problem for these models finding excellent proximity between the two reconstructions of the coupling if the correlation between the oscillators is moderate. Better proximity between the input series makes the reconstructed values more remote. In this paper, we reconstruct the coupling between the solar hemispheres with the two models using the solar faculae as the input series. The reconstructed coupling is weak, in line with results of Norton and Gallagher [39].

The reconstruction with the Kuramoto model is obtained as the rigorous solution of the inverse problem proposed and efficiently applied to solar proxies by Blanter et al. [32]. The inverse problem for the VdP coupled oscillators is solved numerically. As a result, we obtain the coupling between solar hemispheres from the two models. Whilst the overall dynamics of reconstructed couplings qualitatively coincides, their difference exhibits significant variations, Figure 4. The proximity between the two reconstructions is measured with the function $r_{\Delta \omega }\left(t\right)$, defined by (Equation 14), which would be equal to 1 if the reconstructions were identical, Figure 4. This function $r_{\Delta \omega }\left(t\right)$ varies slowly in time and exhibits a systematic bias towards the values that are a bit lesser than 1. Based on [25], we argue that the level of the correlation between the solar faculae series is larger than that required for the most similar reconstructions.

In this paper, VdP oscillations are considered as a (huge) simplification of the MHD-based models [26,27]. The proximity of both reconstructions links the efficient reconstruction of the coupling and phase differences between solar proxies performed by Blanter et al. [11,32] with the ad hoc Kuramoto models to the solar dynamo models inferred from the MHD-equations [2].

In general, arising long-term asymmetry in the solar hemispheres has been vastly investigated empirically, [5,8,40,41]. Various methods of the time-series analysis, including the Fourier transform, wavelets, singular spectrum analysis, and cross-recurrence plot are used to derive the periodicities and trends of the absolute (N−S) or normalized (N−S)/(N+S) asymmetries, where *N* and *S* represent the magnetic fields associated with the northern and southern hemispheres respectively. Nevertheless, only a few authors focused on the phase difference; the paper by Blanter et al. [11] gives a possible example. Complementing this paper, we end up with the conjecture that the solar hemispheres more rapidly diverge than converge, Figure 6.

We estimate the dynamics of the phase difference between the northern and southern hemispheres of the Sun (Figure 6). It is possible that the oscillators move apart more quickly than tend to each other. A general decrease observed from 1919 to 1958 including ∼2-years abrupt growth in the restored phase difference may be related to the traces of 30-to-40-years periodicities found with harmonic analysis in the amplitude characteristics of solar activity (e.g. [42,43,44]). Blanter et al. [45] found the same 3 cycle quasi-periodicities when modeling coupled oscillations between solar proxies with the Kuramoto equations used also in this paper. Two downward trends of the reconstructed normalized frequencies to moderate minima, which are attained in the 1940s and 1990s, were altered by a subsequent growth (Figure 6). In contrast, a deep minimum assigned to the 1950s occurred prior to the episode of the de-synchronization between the two hemispheres. The last at the right restored values of the normalized frequencies continue a downward trend but still remain larger than that of the 1950s. Only additional values can give evidence of how far the series is from a new episode of the de-synchronization.

We stress that the two principally different models usually agree with each other. This agreement is in favor of the reliability of the models. Roughly speaking, the epoch of the synchronized oscillators can be described not only with the MHD but also with simple ad-hoc models. However, both our models lose their adequacy when tackling cycle 20 characterized by the de-synchronization of the polar faculae. The failure of the models is predictable, as they are built on the assumption of the synchronization between the considered time series. However, the divergence between the models dealing with the de-synchronized data rises important questions regarding the nature of the physical mechanisms which stay beyond the model divergence. As such divergence stems from the different origins of nonlinearity (we work with nonlinearly coupled linear model and linearly coupled nonlinear model), further investigation of such simple models could shed light on the adequate choice of MHD parameters. As a result, a better understanding of the relationship between non-linear characteristics of our models and instabilities in the solar dynamo evolution opens the door to the prediction of these instabilities in advance.

The polar faculae were chosen in this paper as available long daily time series representing the polar magnetic fields. The computational procedure can be repeated with shorter time series exhibiting the components of the solar magnetic fields or sunspot data in order to compare the results.

In the same time, the numerical procedure establishing the relationship between the coupling, the phase difference, and the correlation of the VdP oscillators is useful. It functions efficiently when the variables exhaust a synchronization domain between the boundaries Δω=0 and Δω=μ corresponding to the absolute and disappearing synchronization respectively. The numerical analysis is worth being extended to the case of three coupled VdP oscillators and applied to the reconstruction of the velocity profile of the meridional flow. This approach will connect the results by Blanter et al. [28] inferred from the properties of the Kuramoto oscillators to that by Hazra et al. [29], Featherstone and Miesh [46], Choudhuri [47], Cameron et al. [48] obtained with elaborated versions of the solar dynamo model.

## Figures and Tables

**Figure 1 entropy-22-00945-f001:**
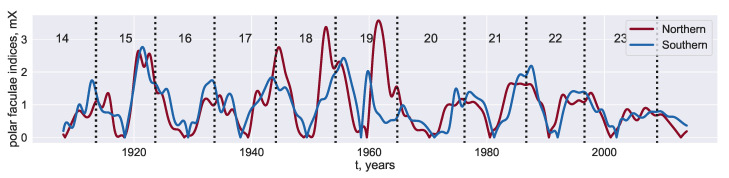
Polar faculae data for northern (blue curve) and southern (red curve) hemispheres.

**Figure 2 entropy-22-00945-f002:**
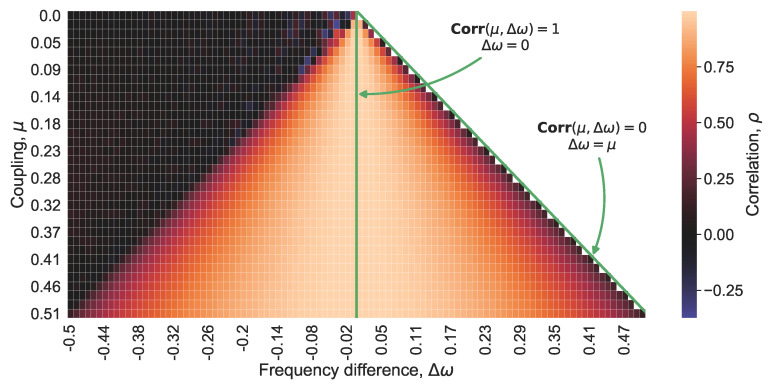
The relationship between coupling μ, Δω, and the correlation ρ between two coupled van der Pol oscillators; Δω<μ.

**Figure 3 entropy-22-00945-f003:**
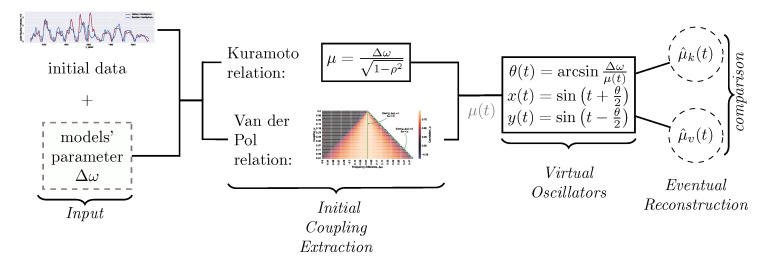
Reconstruction scheme described in details in steps (A)–(C).

**Figure 4 entropy-22-00945-f004:**
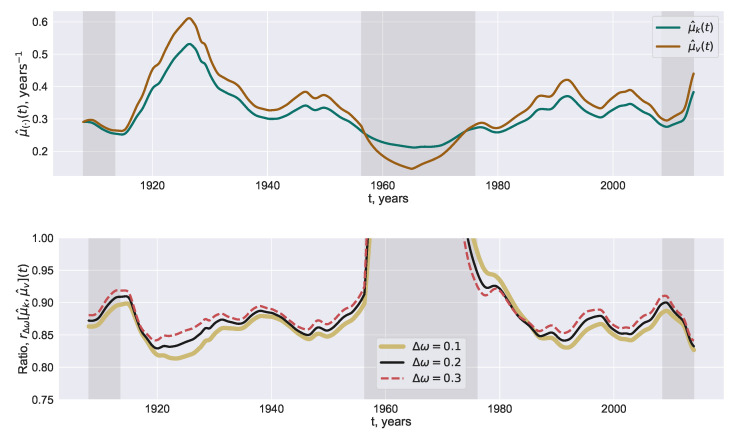
**Top panel**: Reconstructed coupling with the Kuramoto (${\hat{\mu }}_{k}\left(t\right)$; green) and van der Pol (${\hat{\mu }}_{v}\left(t\right)$; orange) models found with Δω=0.2. **Bottom panel**: ratio $r_{\Delta \omega }\left[{\hat{\mu }}_{k},{\hat{\mu }}_{v}\right]\left(t\right)$ defined by (Equation 14) exhibiting the proximity of the reconstructions. Grey figure background means that the reconstruction of the coupling involves either negative correlation between the series (in the middle) or the computation of the correlation on smaller windows (at the left and right); Δω=0.1,0.2,0.3.

**Figure 5 entropy-22-00945-f005:**
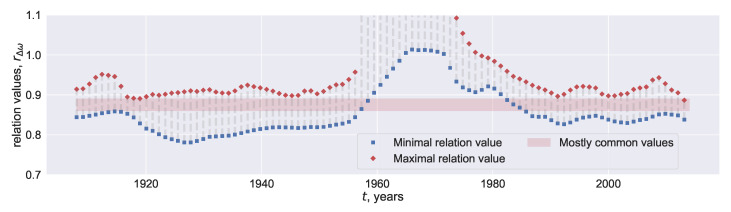
The minimal (blue) and maximal (red curve) values of $r_{\Delta \omega }\left(t\right)$ obtained with different Δω vs. time.

**Figure 6 entropy-22-00945-f006:**
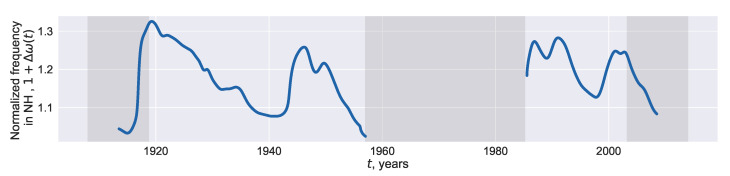
Reconstructed normalized natural frequency, 1+Δω, associated with the signals coming from the northern solar hemisphere. The reconstruction rule is inferred from the assumption that the Kuramoto and VdP reconstruction of the coupling result in the time-independent values of $r_{\Delta \omega }\left(t\right)$ fixed to 0.86.

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
