# Peer review of "Dynamics of Phase Synchronization between Solar Polar Magnetic Fields Assessed with Van Der Pol and Kuramoto Models"

_entropy, 2020, doi:10.3390/e22090945_

Round 1

Reviewer 1 Report

The authors established the similarity in two model-based reconstructions of the coupling between the polar magnetic fields of the Sun represented by the solar faculae series by using the Van der Pol and Kuramoto equations. The manuscript fits very well with the Entropy topics and it is quite well written (I suggest to check some typos), it is interesting and it overall suitable for publication. My recommendation is to review some aspects and to provide more explanation as reported below.

  1. The authors report the analysis of the solar faculae series. Would be interesting to compare the results by using hemispheric magnetic field data? Since both models can be seen as a substantial simplification of solar dynamo models, how the models, and the phenomenon of synchronization, would work with magnetic field data?
  2. The evolution of the phase difference between the polar fields seems to suggest that they tend to move apart more quickly than approach each other. Could this be compared with MHD dynamo models?
  3. How the parameters of both models can be related to physical quantities, and especially, how they have been chosen? The appearance of faculae is obviously related to the Schwabe 11-yr solar cycle but the polarity of the solar magnetic field evolves on the 22-yr cycle. Is this taken into account?

Reviewer 2 Report

The paper entitled “Dynamics of Phase Synchronization between Solar Polar Magnetic Fields Assessed with Van der Pol and Kuramoto models” introduces a measure of similarity in the reconstructions of the coupling performed with Kuramoto (which is a ad-hoc model) and VdP system of two oscillators. The main engine of the paper is to leverage previously reported analytic results on a closed form/table-like relationship between statistical correlation to the system parameter.

This paper is well-written and looks interesting to the community of phase synchronization dynamics in the solar physics. However, I want the authors to address the following questions before I can recommend this paper:

On the technical part:

  1. Can the authors elaborate on the sentence below equation 8, what does it mean for information about what an oscillator leads is lost? It would be nice to show an example.

  2. When the authors said the model equations are fully synchronized, this is because the phase difference would converge to a fixed point as the constant forcing is not strong enough to let it jump out of the attractor. The authors might want to make it bit more clear to other readers who are not familiar with the equations.

  3. When the author says the rule 8 is also time-dependent, this is simply because the \rho varies as a function of time, so the authors might want to explicitly state rho(t) to make easier for readers.

  4. The equation 6 is actually a bit weird because d/dt theta_x – d/dt theta_y in the solar proxies model only synchronized at t equals infinity. So the author might want to state the fact that when \mu > \Delta \omega, let’s define q = theta_x-theta_y, q will actually goes to the fixed point pretty fast, so at large t, one should expect approximation 9 holds as long as one choose to wait for the solar proxies model to run long enough. Thus, this is perhaps not an assumption for solar proxies.

  5. On the figure 3, on the block above virtual oscillator, the first row should be theta(t) = acrsin(\Delta \omega/\mu(t)), not arccos. Same things applies to equation 13.

  6. In section 4.2, the authors said they sweep all possible values of \Delta \omega to find the most frequent r_{\Delta \omega} as a fixed constant. This basically performs an uncertainty propagation of the distribution of \Delta \omega to the nonlinear observation function r_{\Delta \omega}, then find the latter’s most probably location. However, it seems the authors simply assume uniform distribution of \Delta \omega? If so, what’s the range of all possible values of \Delta \omega?

  7. Maybe I missed something, why does the similarity between two ad-hoc model-based reconstructions are important? Since they are both ad-hoc models and one can find arbitrary other models to represent the similar phase sync behavior.

On grammar part:

  1. In abstract, “the both reconstruction procedures” should be “both of the reconstruction procedures”

  2. Caption in Figure 1 says black curve vs green curve but in the figure it is red vs blue curve.

  3. In the conflicts of interests, it should be “the” rather than “he”

  4. In figure 4, it is not pink figure background, it seems grey background. Also, is the correlation becomes larger than 1? it doesn’t look like negative correlation.

Reviewer 3 Report

In this paper, the dynamics of phase synchronization between solar
polar magnetic fields assessed with van der pol and Kuramoto models
are investigated. The discussions could be interesting and the results could be useful for the related future works. Hence, if the following points are reconsidered, this paper could be worthy of being published.

1. There would exist the past related works on the dynamics of phase synchronization between solar polar magnetic fields in the literature. By comparing with these preceding studies, the new ingredients and significant progresses of this work should be stated more explicitly and in more detail. That is, the differences between this paper and the past ones should be described in more detail and more clearly.

2. How sensitive do the interpretations of the results depend of the framework of the models such as van der pol and Kuramoto models?

3. From the considerations of the reconstruction procedures and the
8 evolution of the phase difference between the polar fields, what can we learn on fundamental physics in the sun?

4. It is finally recommended that the wordings and grammar of English should be rechecked throughout the present manuscript.
